# MULTI-SCALE FUSION SELF ATTENTION MECHANISM

## ABSTRACT

Self attention is widely used in various tasks because it can directly calculate the dependency between words, regardless of distance. However, the existing self attention lacks the ability to extract phrase level information. This is because the self attention only considers the one-to-one relationship between words and ignores the one-to-many relationship between words and phrases. Consequently, we design a multi-scale fusion self attention model for phrase information to resolve the above issues. Based on the traditional attention mechanism, multi-scale fusion self attention extracts phrase information at different scales by setting convolution kernels at different levels, and calculates the corresponding attention matrix at different scales, so that the model can better extract phrase level information. Compared with the traditional self attention model, we also designed a unique attention matrix sparsity strategy to better select the information that the model needs to pay attention to, so that our model can be more effective. Experimental results show that our model is superior to the existing baseline model in relation extraction task and GLUE task.

## 1 INTRODUCTION

Attention mechanism is a model widely used in natural language processing tasks. Attention determines where the model needs attention by constructing an attention matrix. With the in-depth study of attention model, Vaswani et al. (2017) proposed a more advanced self attention mechanism. The self attention model dynamically constructs the attention matrix by calculating the correlation degree between words. Compared with the traditional attention mechanism, the self attention model can construct different attention matrices for different inputs and retain more information.

Although the current self-attention model has achieved relatively successful results, we have found a serious problem, that is, self-attention can directly calculate dependencies between words. Although this attention matrix can pay good attention to the relevant information between one-to-one words. However, in real life, the language environment is very complicated. There are often many phrases in sentences. These phrases may contain many words, but the words that make up the phrase often cannot fully express the meaning of the phrase itself. At this time, we need to be able to extract the one-to-many relationship between words and phrases. for instance:

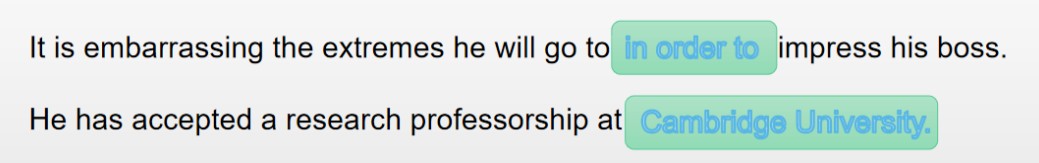

Figure 1: Examples of phrases in sentences.

In Figure 1, we give two examples. The two parts framed in green in the sentence are two phrases in the sentence. The meaning expressed by phrases such as "in order to" and "Cambridge University" in the two sentences can not get the corresponding meaning from any word of the two phrases. Therefore, these phrases need to be regarded as a whole to construct the attention matrix, so as to ensure that the information extracted by the model is correct. In previous research work, Yao et al. (2019) conducted pooling operation on entity words in relation extraction task to achieve the effect

of phrase information coding. However, this operation has great limitations, that is, we need to know the position and length of phrases in advance, which is very difficult for general tasks.

In this paper, we propose to use convolution kernels of different sizes to extract phrase level information in sentences, and learn whether to interact with the sampled information adaptively by the model itself. At the same time, not all scale information is required by the model, and the conflict between phrase information and its constituent words should be taken into account when selecting. For example, as shown in sentence 1 of Figure 1, when calculating the weight of attention matrix, if the relevance of (go, in order to) is large, the attention relevance of (go, in) (go, order) (go, to) should be reduced accordingly, and vice versa. Therefore, we design a unique matrix sparsity strategy, which can better adapt to our multi-scale fusion self-attention model.

The experimental results show that our model has a better effect on the relationship extraction task, and achieves a better level than the baseline on the GLUE data set.

The main contributions of this paper include:

1. Based on the traditional self attention mechanism, the phrase level representation is extracted through sampling at different scales, and the attention matrix is constructed by using the representation, which improves the deficiency that the attention model can only extract one-to-one information between words.

2. On the basis of integrating multi-scale information, in order to better guide the model for information selection, a sparsity strategy of attention matrix is proposed, which can better select the information that needs to be focused when constructing attention matrix.

## 2 RELATED WORK

The initial attention mechanism is widely used in natural language processing tasks as a model to integrate information. The initial attention mechanism is often used as a model to learn the association between hidden vectors after Recurrent Neural Network (RNN). Note that the emergence of the model breaks the limitation that the traditional encoder decoder structure depends on an internal fixed length vector during encoding and decoding (Zhou et al. (2016))

Du et al. (2018) Found in many experiments that the original single one-dimensional vector can no longer meet the requirements of extracting information diversity, so they proposed to build a 2-D attention matrix to adapt to more complex situations. Among them, they believe that each dimension in such a 2-D level attention matrix represents a different focus direction

Vaswani et al. (2017) put forward the self attention mechanism in the transformer model. The author takes the attention model as the main structure of the model, improves the parallelism of the model, and changes the invariable characteristics of the previous attention matrix. The self attention model creates more contextual representations by designing different attention matrices for different samples.

Correia et al. (2019) considered that the complexity of the model will increase with the increase of sentence length in the operation process of self attention mechanism, which makes it difficult for the model to deal with long text information. Therefore, the author puts forward some suggestions $\alpha$- Entmax method to construct a sparse attention matrix. By constructing such a sparse attention matrix model, the parameters are reduced, which can adapt to longer text data.

Li et al. (2021) aiming at the problem of insufficient information extraction in the traditional attention mechanism, proposed a method of using the hard attention mechanism to extract the important information of the model. At the same time, the author also proposed a method of adding negative information to the attention matrix to enrich the diversity of the model.

### 2.1 SELF ATTENTION

Before introducing our work, we first briefly introduce the self attention model of Vaswani et al. (2017) in transformer.

Formally, We assume that H is a vectorized representation of a sentence with length n. The attention mechanism starts by projecting the tokens into three subspaces: the query subspace Q, key subspace

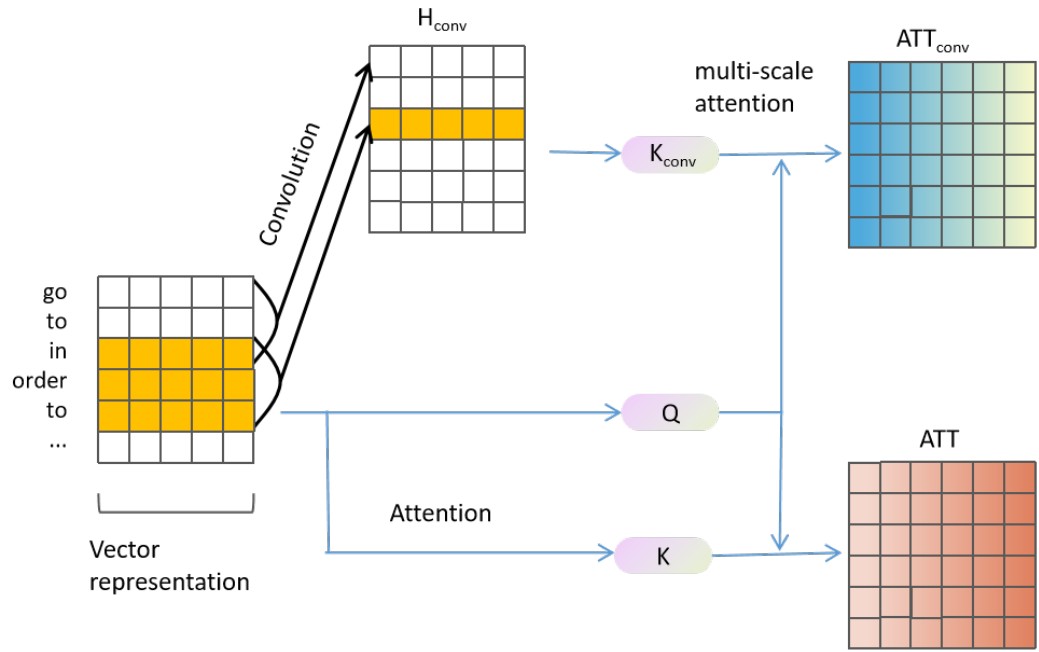

Figure 2: General architecture for multi-scale attention module.

K, and value subspace V, with corresponding projection matrices $W^Q$, $W^K$, $W^V$. It then computes the n × n attention matrix A as follows.

$$A = softmax((Q \cdot K^T)/\sqrt{d_k}) \tag{1}$$

Where $d_k$ is a scaling factor, T represents transpose operation, and K is used to calculate the correlation score between each item and the query. Then, the normalized attention weight is calculated using softmax, and the obtained attention matrix A is used to weight the value of each item in each query context. Finally, the obtained attention matrix A is multiplied by vector V to obtain the final sentence representation.

## 3 METHOD

In this section, we propose a general attention model, multi-scale fusion self attention mechanism, which can pay attention to the phrase level information in the sample. Our multi-scale fusion self attention model consists of two parts: multi-scale attention module and dynamic sparse module. After obtaining the corresponding coded representation of each word of the input sentence, the multi-scale attention module obtains the attention matrix at different scales. The dynamic sparse module constructs a dynamic coefficient strategy to construct a sparse attention matrix according to the summarized attention characteristics. Finally, the constructed attention matrix is multiplied by the value matrix to obtain a sentence representation with phrase level information.The multi-scale attention module is shown in Figure 2.

### 3.1 MULTI-SCALE ATTENTION MODULE

Given a sentence S of length n, where $S_i$ (1≤i≤n) is the word in the sentence, we can get their vector representation after encoding. The encoder can be Bert or LSTM. Take Bert as an example:

$$H = Bert(S) \tag{2}$$

Where H ∈ $R^{n*d}$ is the vectorized representation of the sentence, where d is the encoding length, and $H_i$ is the encoding representation of each words in the sentence. In order to obtain vector representation of different scales, we use convolutional neural network to further extract information from

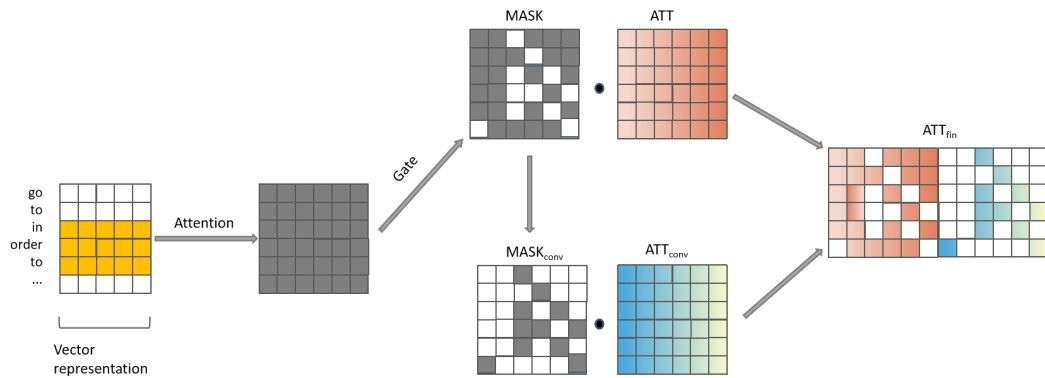

Figure 3: General architecture for dynamic sparse module.

sentence representation. Convolutional neural network (CNN) cannot integrate global information because it is limited by the scale of receptive field of convolution kernel, but it also makes a great improvement for Convolutional neural network to extract local informations, We use this to control the length of phrase level information to be extracted by setting the scale k of convolution kernel, so as to achieve the effect of integrating phrase level information.

Convolution is an operation between the weight vector $w_c$ and the input vector of the input sequence H. The weight matrix $w_c$ is regarded as a convolution filter. We assume that the length of the filter is k, so $w_c \in R^{k*d}$. Convolution involves multiplying the dot product of $w_c$ by each k-gram in sequence H to obtain another sequence $H_{conv}$ :

$$H_{conv}^j = w_c \odot H_{j-k+1:j} \tag{3}$$

Where $\odot$ means multiply element by element. In order to ensure that the length of $H_{conv}$ is consistent with H, we padded the convolution operation. At the same time, due to the ability to capture different features, it is usually necessary to use multiple filters (or feature maps) in convolution. Therefore, we use d filters ($W_c = w_c^1, w_c^2, \cdots, w_c^d$), and the convolution operation can be expressed as:

$$H_{conv}^{ij} = w_c^i \odot H_{j-k+1:j} (0 \le i < d) \tag{4}$$

The convolution result is matrix $H_{conv} = [H_{conv}^1, H_{conv}^2, \cdots, H_{conv}^d] \in R^{d*n}$. H integrates the phrase level information with K as the length. In the process of constructing the attention matrix, we refer to the construction method of Vaswani et al, and make corresponding modifications according to the characteristics of our task. The formula is as follows:

$$Q = H \cdot W^Q \tag{5}$$

$$K = H \cdot W^K \tag{6}$$

$$K_{conv} = H_{conv}^T \cdot W^K \tag{7}$$

$$ATT = softmax((Q \cdot K^T)/\sqrt{d_k}) \tag{8}$$

$$ATT_{conv} = softmax((Q \cdot K_{conv}^T)/\sqrt{d_k}) \tag{9}$$

The above formula is our method to construct a multi-scale attention matrix, where $W^Q$ and $W^K$ are trainable parameter matrixs with the shape of d*d. $K_{conv}$ is the key matrix calculated by $H_{conv}$ integrating phrase level information. The difference from the construction of Vaswani et al is that we calculated $ATT_{conv}$, which is obtained by multiplying the query matrix with word as the minimum granularity and the key matrix $K_{conv}$ with phrase as the minimum granularity. Its purpose is to calculate the degree of attention between words and phrases on different scales.

## 3.2 DYNAMIC SPARSE MODULE

In constructing the multi-scale attention model, we note that although there is phrase information in sentences, not all phrases with length k can be expressed as phrases. At the same time, we should also consider the differences between the traditional attention matrix and the multi-scale attention

matrix, that is, if a word chooses to pay attention to a phrase, it should reduce the attention to each word in the phrase. Therefore, we design a dynamic sparse module for our multi-scale attention module, which can be dynamically adjusted to ensure that the model can select more suitable phrase information.The dynamic sparse module is shown in Figure 3. The method is as follows.

$$Q_1 = H \cdot W^{Q1} \tag{10}$$

$$K_1 = H \cdot W^{K1} \tag{11}$$

$$ATT_{cut} = softmax((Q_1 \cdot K_1^T)/\sqrt{d_k}) \tag{12}$$

$W^{Q1}$ and $W^{K1}$ are two trainable parameter matrices with the shape of d * d. in the above formula, we choose to reconstruct a network for calculating similarity. Different from formula 8, this formula will be used to construct a sparse attention matrix. We set a threshold $\mu$ for att matrix to select whether the information is truncated. If the value of this position is greater than the threshold, it will be set to 1. If not, it will be set to 0. The formula is as follows:

$$Mask^{qj} = 1 \quad if(ATT_{cut}^{qj} > \mu) \quad else \quad 0 \tag{13}$$

Because the size of the threshold will closely affect the location of the matrix's attention, here we set the threshold to 1 / n, so we get a mask matrix composed of 0 and 1. Because this mask matrix only uses word level attention, we also need to build a mask matrix of multi-scale attention matrix. In order to realize the difference between traditional attention matrix and multi-scale attention matrix, we choose to reverse build the mask matrix of multi-scale attention matrix according to Mask matrix. The formula is as follows:

$$Mask_{conv} = E - Mask \tag{14}$$

Where E is a matrix with all values of one, so we can obtain the corresponding sparse multi-scale attention matrix according to the two mask matrices. The formula is as follows:

$$ATT_M = ATT \odot Mask \tag{15}$$

$$ATT_{Mc} = ATT_{conv} \odot Mask_{conv} \tag{16}$$

$$ATT_{fin} = [ATT_M, ATT_{Mc}] \tag{17}$$

$ATT_{fin}$ is the final attention matrix, which is obtained by splicing the traditional attention matrix and our multi-scale attention matrix in the last dimension,so it's shape is n * 2n. $ATT_{fin}$ represents the correlation degree between the i-th word and the j-th word. When $1 \leq j \leq n$, its value represents the attention value between words, and when $n < j \leq 2n$, it represents the attention value between the i-th word and the (j-n)-th phrase.

$$V = \begin{bmatrix} H \\ H_{conv}^T \end{bmatrix} \cdot W^V \tag{18}$$

$$S = ATT_{fin} \cdot V \tag{19}$$

The calculation method of the value matrix is shown in formula 16. For the trainable parameter matrix with the shape of d * d at $W^V$, we splice H and $H_{conv}$ in the first dimension and obtain the value matrix V with the shape of 2n * d through linear transformation.Finally, we multiply the obtained attention matrix $ATT_{fin}$ and value matrix V to obtain the final sentence representation. After obtaining the representation, we can use pooling or other dimensionality reduction methods to realize the final flat representation.

## 4 EXPERIMENT AND EVALUATION

We evaluated the models of multiple benchmark data sets under multiple tasks. Experiments show that our multi-scale self attention mechanism achieves better results than the traditional self attention mechanism in sentence level relationship extraction task and GLUE task.

### 4.1 RELATION EXTRACTION

Traditional methods usually deal with relationship extraction task (RE) in supervised learning mode to extract the relationship between two entities mentioned in a sentence. Each sentence begins by manually labeling two entities. Then a model is needed to predict the relationship between these

| SYS | SemEval | Wiki80 |
|---|---|---|
| CNN(Han et al. (2019)) | 71.1 | 63.9 |
| TRE(Alt et al. (2019)) | 87.1 | - |
| SpanRel(Jiang et al. (2019)) | 87.4 | - |
| BERT(Han et al. (2019)) | 88.0 | 84.6 |
| BERT-Entity(Soares et al. (2019)) | 88.3 | 86.6 |
| $AR_{semeval10,0}$(Zhu et al. (2020)) | 88.5 | - |
| Han et al. (2020) | - | 86.1 |
| BERT-MS(our model) | **88.9** | **87.0** |

Table 1: F1 value in SemEval 2010 task 8 and AUC value in Wiki80.

| Corpus | Train | Test | Task | Metrics | Domain |
|---|---|---|---|---|---|
| Single-Sentence Tasks | | | | | |
| CoLA | 8.5K | 1K | acceptability | Matthews corr | misc |
| SST-2 | 67k | 1.8k | sentiment | acc. | movie reviews |
| Similarity and Paraphrase Tasks | | | | | |
| MRPC | 3.7k | 1.7k | paraphrase | F1 | news |
| QQP | 364k | 391k | paraphrase | F1 | social QA questions |
| Inference Tasks | | | | | |
| MNLI | 393k | 20k | NLI matched | acc./mismatched acc. | misc. |
| QNLI | 105k | 5.4k | QA/NLI | acc. | Wikipedia |
| RTE | 2.5k | 3k | NLI | acc. | news, Wikipedia |

Table 2: Task descriptions and statistics. All tasks are single sentence or sentence pair classification.

| SYS | MNLI-(m/mm) | QNLI | SST-2 | CoLA | RTE | QQP | MRPC |
|---|---|---|---|---|---|---|---|
| BiLSTM+CoVe | 65.4/65.7 | 70.8 | 81.9 | 18.5 | 52.7 | 84.9 | 78.7 |
| BiLSTM+CoVe+Attn | 68.1/68.6 | 72.9 | 80.7 | 8.3 | 56.0 | 83.4 | 80.0 |
| BiLSTM+ELMo+Attn | 76.4/76.1 | 79.8 | 90.4 | 36.0 | 56.8 | 84.3 | 84.4 |
| OpenAI GPT | 82.1/81.4 | 87.4 | 91.3 | 45.4 | 56.0 | 70.3 | 82.3 |
| Nystromformer | 80.9/82.2 | 88.7 | 91.4 | - | - | 86.3 | 88.1 |
| BERT-base | **83.9**/84.1 | 90.6 | 92.3 | 56.5 | **65.7** | 90.7 | 88.8 |
| CHARFORMER$_{Tall}$ | 83.7/84.4 | 91.0 | 91.5 | 51.8 | - | **91.4** | 91.4 |
| BERT-MS | 83.6/**84.4** | **91.4** | **93.1** | **58.5** | **65.7** | 91.0 | **91.5** |

Table 3: Results on GLUE tasks. We report F1 score for MRPC and QQP and accuracy for others.

annotation entities. As there are many efforts to adopt models for this setting (Zeng et al. (2014); Zhang et al. (2015); Lin et al. (2016)) In order to verify the effectiveness of our method, we selected two relational extracted data sets for experiments, namely semeval 2010 task 8 (Hendrickx et al. (2010)) and wiki80. Semeval 2010 task 8 dataset contains 10 distinguishable relationships (causality, tool agent, product producer, content container, entity source, entity destination, component whole, member set, message subject, and others). The first nine relationships have two directions, while the remaining relationships have no direction, so the total number of relationships is 19. The data set consisted of 10717 annotated sentences with 8000 and 2717 training and test samples, respectively.

Wiki80 comes from fewrel (Han et al. (2018)), which is a large-scale few shot data set. It contains 80 relationships and 56000 instances from Wikipedia and Wiki data. Since Wiki80 is not an official benchmark, we directly report the results on the validation set.

In the experiment, we choose Bert as the encoder of our model. The experimental results show that our model is superior to the existing limit model in performance. Table 1 shows the experimental results of the model on semeval 2010 task 8 and Wiki80 data set.

From the Table 1, we can see that our model is written as "BERT-MS". Because we embed our model as a fine-tuning model behind the Bert model, that is, after the sentence is pre trained, the language model obtains the vector representation of each word, and then connected to our multi-scale self-attention model, so as to obtain the sentence representation that can extract phrase level

information by using the information of different scales. The next step is to get the flat sentence representation through polling or other dimensionality reduction methods, and finally get the final classification result through a fully connected neural network. Compared with the traditional Bert model, we only add a multi-scale self-attention model layer after the Bert model, and our model does not make any other changes. Similarly, the model in subsequent experiments also follows this model architecture.

Compared with the previous methods that only use entity information Soares et al. (2019) or pool entity and context respectively as the guidance information of relationship classification Zhu et al. (2020), our multi-scale self-attention mechanism can better integrate the importance of entity words and context information in sentences, and better extract the interactive information between entity words and phrase in sentences, So as to improve the accuracy of model classification.

In the experiment, we use Adam optimizer to optimize the model, and the learning rate is 3e-5. It can be seen from the results that our model can better adjust the Bert model to obtain better results than the baseline.

## 4.2 GLUE

GLUE The General Language Understanding Evaluation (GLUE) benchmark is a collective task for multiple natural language understanding. The glue benchmark includes the data sets shown in Table 2, and the descriptions of these datasets were initially summarized by Wang et al. (2018)

We used Bert-base as the encoder of our model. The batch size is set to 32 and the appropriate learning rate is selected in (5e-5, 4e-5, 3e-5 and 2e-5). The final results are shown in Table 2.

We compare the Nystromformer model of Xiong et al. (2021), CHARFORMER$_{Tall}$ model of Tay et al. (2021)and BERT-base model. Compared with the network constructed by character level information and block information composed of characters (Tay et al. (2021)), our model selects the network constructed by word and phrase information, which can better find the hidden information in sentences and has stronger interpretability. Due to the existing baseline model, it can be determined that our multi-scale self-attention model can integrate the information of different scales, so as to achieve superior experimental results.

## 4.3 SELECTION OF K VALUE

In the above, our experiments are based on k = 3, that is, the model can only extract phrase level information with length of 3. In this section, we will conduct experiments for different k values to verify the impact of k value on the performance of the model. We chose semeval 2010 task 8 to validate our experiment. Table 4 shows our experimental results. In the experiment, we use the

| k | SemEval |
|---|---------|
| k=3 | 88.9 |
| k=5 | 88.6 |
| k=7 | 88.4 |
| k=3,k=5 | **89.1** |
| k=5,k=7 | 88.7 |

Table 4: F1 value of different k value.

fusion method for phrase information with different lengths, that is, if we choose (k = 3, k = 5), it means that we need to calculate the convolution information with 3 as the convolution kernel size and 5 as the convolution kernel size at the same time. In the sparse strategy, the two choose the same sparse matrix, and then the model chooses which scale should pay more attention to. It can be seen from the table that when (k = 3, k = 5), the model can get good experimental results on semeval 2010 task 8, which shows that more phrase level information in the sentence is generally concentrated in these two lengths. When k = 7, the effect is not good, which also shows that there are few phrases of this length in this data set. This experiment does not mean that using the same strategy in all tasks can get the best effect. Different data sets have different adaptive k values, which need to be adjusted according to the characteristics of the data set to achieve the best experimental effect.

## 5 CONCLUSION

We improve the existing self attention mechanism and propose a self attention mechanism that can integrate phrase level information, which overcomes the disadvantage that the traditional attention mechanism can not effectively extract phrase level information. Due to the universality of this method, it can be extended to other application scenarios of self attention mechanism, such as translation tasks. Future work will include optimizing our model and understanding how to select features to improve its performance.

## ACKNOWLEDGE

This work was supported by the Innovation Foundation of Science and Technology of Dalian under Grant No.2018J12GX045 and National Key R&D Program of China under Grant No.2018AAA0100300.

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

## A  APPENDIX

### A.1  ATTENTION VISUALIZATION

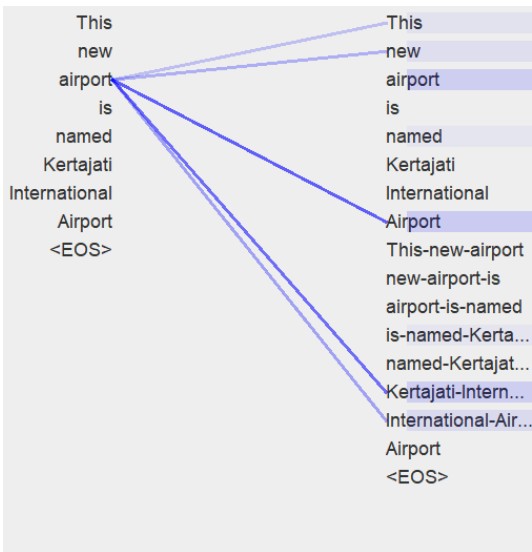

Figure 4: An example of the attention mechanism, the darker the line color in the figure, the greater the value of attention. We can see that our model successfully pays attention to the phrase level information of "Kertajati International Airport"

