# OpenReview forum: "Multi-scale fusion self attention mechanism"
_ICLR.cc/2022/Conference — ICLR 2022 Submitted_

### Official Review · Reviewer_c4sd · 2021-11-02

**Correctness:** 3
**Technical Novelty And Significance:** 2
**Empirical Novelty And Significance:** 2
**Recommendation:** 3
**Confidence:** 3

**Main Review:**

I think the novelty is limited. Using convolution operation with different kernel sizes to extract information of different scales is quite general, e.g., googlenet [1]. It seems it merely transfers the idea from multi-scale context information modeling on some vision tasks to sequential modeling tasks.
Besides, I think simply convolution operation with different kernel sizes can help to extract contextualized information. However, it seems it can not guarantee extract phase information.
As claimed in the paper, the dynamic sparse module is proposed to realize the end that if a word chooses to pay attention to a phrase, it should reduce the attention to each word in the phrase. However, this work simply sets the attentional values that are smaller than a threshold to 0 to realize this end. I do not why this mechanism can realize the above-mentioned end. Besides, the contribution of this point is not verified.

[1] Szegedy et al., Going deeper with convolutions, in CVPR, 2015.

**Summary Of The Paper:**

This work proposes a multi-scale fusion self-attention module to help extract phase-level at different scales. It utilizes convolution operations with kernels of different sizes to achieve this end. The authors conduct experiments on the relation extraction and GLUE tasks to demonstrate its effectiveness.

**Summary Of The Review:**

I think the novelty is limited and some explanation is not clear.

---

### Official Review · Reviewer_iGQt · 2021-11-02

**Correctness:** 3
**Technical Novelty And Significance:** 2
**Empirical Novelty And Significance:** 2
**Recommendation:** 3
**Confidence:** 3

**Main Review:**

Despite being motivated by the attention between phrase and word, the empirical and technical novelty is limited.
1. The proposed method essentially adds additional transformations (and therefore parameters) in the transformer architecture, so the performance improvement is expected. It is unclear whether the proposed method is indeed better when it has the same number of parameters and operations as the baselines.
2. The idea of combining convolution and transformer architecture has been widely adopted in prior works.
3. It is not clear whether modeling phrases explicitly is indeed helpful, especially in a deep neural network. The model may already capture the phrases implicitly.

To improve the paper, the authors should try to
1. Verify that the proposed architecture is indeed superior to alternative design when having similar number of parameters and operations (e.g. what's the performance w.r.t model size, can the proposed model be applied to the backbone network to achieve better performance?)
2. Show that the superior performance is not simply the result of more parameters (e.g. what happens if we set k=1 and increase the number of filters)
3. Show evidence about why modeling phrases explicitly is important and the proposed method indeed capture the phrase information
4. Explain why existing model is not sufficient for capturing phrase information


**Summary Of The Paper:**

This paper introduces a new network architecture based on a transformer. The basic idea includes
1) Extract phrase information using a convolution operator and compute the attention between phrases and words
2) Learn to predict a mask for word-word attention to turn off word-phrase attention when word-word attention is high
Empirical results show that the proposed method improves the baseline BERT model when being applied to the features extracted by BERT.


**Summary Of The Review:**

This paper falls short of showing the benefit of the proposed method. It is unclear whether the new architecture is indeed better than existing alternatives both theoretically and empirically.

---

### Official Review · Reviewer_hsJZ · 2021-11-02

**Correctness:** 3
**Technical Novelty And Significance:** 2
**Empirical Novelty And Significance:** 2
**Recommendation:** 3
**Confidence:** 4

**Main Review:**

The contribution of the manuscript is minor and the novelty of the proposed method is marginal. The motivation of the proposed method is not convincing. There are many existing methods that can extract phrase or context information for NLP tasks.

For multi-scale feature extraction or attention fusion, there are also many existing studies. The proposed method just simply use different convolutional kernels for the task. It does not provide many new insights to the community.

There is not a deep analysis of the existing literature. The introduction and related work section should be completely rewritten. The information provided in the current shape is very shallow. Many state-of-the-art studies are missing.

The experimental results cannot fully support the proposed method. The authors should compare the proposed method on more NLP benchmarks with a comparison with the SOTA methods. For example, the most recent methods used for comparison were published in 2020 in Table 1.

As the proposed method is a generic module. It should be evaluated on various NLP tasks.

**Summary Of The Paper:**

The manuscript presents a multi-scale self-attention method for NLP tasks. The aim is to better extract phrase- and word-level features. The main contribution of the proposed method is to apply different kernel sizes for feature extraction and multi-scale attention fusion. Additionally, a mechanism called dynamic sparse module is applied to adjust the weights of the obtained attention matrix.

**Summary Of The Review:**

Overall, the proposed method is not good enough for publishing in top tier conferences such as ICLR. The contribution and novelty of the proposed multi-scale self-attention method are marginal. The experimental results cannot fully support the proposed approach.

---

### Official Review · Reviewer_cTsu · 2021-11-04

**Correctness:** 2
**Technical Novelty And Significance:** 2
**Empirical Novelty And Significance:** 2
**Recommendation:** 3
**Confidence:** 5

**Main Review:**

Strengths:

A simple method that uses convolutional model to enhance the localness of self-attention and phrase-level learning.

Weaknesses:
* Adding convolution into self-attention and capturing phrase information have been well studied before;
* The proposed dynamic masking strategy has flaws;
* Writing needs improvement;
* Experimental details are not always clear;
* Ablation study is incomplete and comparison should be enhanced.


Details:
1)	Adding convolutional models and capturing phrase information for self-attention has been well explored in the context of machine translation. Take the following two papers as an example:
[1] Yang et al., Convolutional Self-Attention Networks [2] Hao et al., Multi-Granularity Self-Attention for Neural Machine Translation These existing studies reduce the novelty of this paper. Also, a direct comparison with these studies is required.
2)	The masking strategy is essentially another attention layer, but its formulation indicates that no gradient will be back-propagated into W^{Q_1} and W^{K_1} as shown by Eq. 10, 11, 12, 13. The authors don’t explain how to optimize them.
3)	Writing needs improvement. In particularly, logistic should be improved.
4)	How did you implement your convolutional model? Did you apply a vanilla convolution or depth separable convolution, or dynamic convolution?
5)	The proposed model is used for fine-tuning after BERT encoding. One noticeable thing is that, although BERT outputs representation for each word, its encoding is fully contextualized. To large extent, BERT encoding contains sentence-level information rather than simply word-level. The authors need more stronger motivation to tell the necessity of convolution in self-attention.
6)	One ablation is missing. What if you stack another vanilla attention layer for finetuning without convolution? This should be added for comparison.


**Summary Of The Paper:**

This paper incorporates convolutional models into self-attention to explicitly handle word-to-phrase correlation, paired with a sparse masking strategy to balance between word-to-word attention and word-to-phrase attention. The model achieves good performance on GLUE and RE tasks.

**Summary Of The Review:**

In short, the authors propose convolutional model for self-attention, which is lack of novelty. The method also has flaws, and the experiments are not very convincing.

---

### Decision · Program_Chairs · 2022-01-20

**Decision:**

Reject

**Comment:**

This paper proposes a multi-scale fusion self attention model for phrase information, which incorporates convolutional models into self-attention to explicitly handle word-to-phrase correlation. This is paired with a sparse masking strategy to balance between word-to-word attention and word-to-phrase attention. The model achieves good performance on downstream tasks.

While the proposed method is simple and looks effective, reviewers have expressed concerns with lack of novelty (see the suggested missing references), lack of clarity in the experimental details, and unclear writing. Unfortunately, there was no response from the authors, which makes me recommend rejection. We urge the authors to follow the reviewers' suggestions in a future iteration of their work.